# Floquet spin states in OLEDs

S. Jamali[1,6], V. V. Mkhitaryan[1,6], H. Malissa [1], A. Nahlawi[1], H. Popli[1], T. Grünbaum[2], S. Bange [2], S. Milster[2], D. M. Stoltzfus[3], A. E. Leung[4,5], T. A. Darwish[4], P. L. Burn [3], J. M. Lupton [1,2✉] & C. Boehme[1✉]

Electron and hole spins in organic light-emitting diodes constitute prototypical two-level systems for the exploration of the ultrastrong-drive regime of light-matter interactions. Floquet solutions to the time-dependent Hamiltonian of pairs of electron and hole spins reveal that, under non-perturbative resonant drive, when spin-Rabi frequencies become comparable to the Larmor frequencies, hybrid light-matter states emerge that enable dipole-forbidden multi-quantum transitions at integer and fractional g-factors. To probe these phenomena experimentally, we develop an electrically detected magnetic-resonance experiment supporting oscillating driving fields comparable in amplitude to the static field defining the Zeeman splitting; and an organic semiconductor characterized by minimal local hyperfine fields allowing the non-perturbative light-matter interactions to be resolved. The experimental confirmation of the predicted Floquet states under strong-drive conditions demonstrates the presence of hybrid light-matter spin excitations at room temperature. These dressed states are insensitive to power broadening, display Bloch-Siegert-like shifts, and are suggestive of long spin coherence times, implying potential applicability for quantum sensing.

[1] Department of Physics and Astronomy, University of Utah, Salt Lake City, UT 84112, USA. [2] Institut für Experimentelle und Angewandte Physik, Universität Regensburg, 93053 Regensburg, Germany. [3] Centre for Organic Photonics & Electronics, School of Chemistry and Molecular Biosciences, The University of Queensland, Brisbane, QLD 4072, Australia. [4] National Deuteration Facility, Australian Nuclear Science and Technology Organization (ANSTO), Lucas Heights, NSW 2234, Australia. [5] Present address: Scientific Activities Division, European Spallation Source ERIC, Lund 224 84, Sweden. [6] These authors contributed equally: S. Jamali, V. V. Mkhitaryan. ✉email: john.lupton@ur.de; boehme@physics.utah.edu

A spin in a magnetic field is a perfect discrete-level quantum system, for which resonant electromagnetic radiation can drive coherent propagation in a well-controlled perturbative fashion following Rabi's theory[1]. Such propagation is used for magnetic resonance applications in spectroscopy and imaging, where the thermal-equilibrium population of spin states is altered under resonance, inducing an effective magnetization change of nuclear or electronic spins[2]. Experiments are generally performed under the condition that the Zeeman splitting between spin states induced by a static magnetic field is much larger in energy, or frequency, than the Rabi frequency[1]. This weak-drive limit of resonant pumping is described by perturbation theory. Magnetic resonance can also be probed by observables secondary to magnetization, such as the permutation symmetry of pairs of spins of electronic excitations. Such spin-dependent transitions are reflected in luminescence or conductivity in atomic, molecular, or solid-state systems[3,4] in optically or electrically detected magnetic resonance (ODMR, EDMR) spectroscopies[5]. Color centers in crystals, like atomic vacancies in silicon carbide or diamond, are widely used in ODMR-based quantum metrology and quantum-information processing[6], but are limited in one regard: as dipolar and exchange coupling in the spin pair is strong, substantial level splitting arises at zero external field, posing a lower limit on resonance frequency. Such a limitation does not exist for weakly coupled spin-½ charge-carrier pairs which form, for example, by electron transfer in molecular donor-acceptor complexes, where they account for a range of magnetic-field effects[7].

A particularly versatile way to study weakly coupled spin-½ pairs is offered by organic light-emitting diodes (OLEDs), which generate light from the recombination of electrically injected electrons and holes that bind in pairs of singlet or triplet permutation symmetry[8]. As the formation rates of intramolecular excitons from intermolecular electron-hole carrier pairs differ for singlet and triplet spin permutations, and singlet pairs tend to have shorter lifetimes than triplets, an increase of singlet content in the electron-hole pair depletes the reservoir of available carriers and leads directly to a change in conductivity[9–11]. Because carrier spins interact with local hyperfine fields, which originate from unresolved hyperfine coupling between charge-carrier spins and the nuclear spins of the ubiquitous protons[12], carrier migration through the active OLED layer gives rise to spin precession and, ultimately, mixing of singlet and triplet carrier-pair configurations. As a result, OLEDs can exhibit magnetoresistance on nanotesla scales at room temperature[5]. A static magnetic field tends to partially suppress this hyperfine spin mixing, an effect that is reversed under magnetic resonance conditions, which give rise to distinct resonances in the magnetoresistance functionality[13].

It is important to stress that such studies are only possible in materials characterized by very weak spin-orbit coupling[14], such as organic semiconductors, and are not generic to electron-hole recombination in LEDs as a whole. In an inorganic LED, for example, spin mixing occurs by spin-orbit coupling in addition to the fact that recombination does usually not occur into tightly bound, and hence thermally stable, excitonic species. Light generation in inorganic LEDs, in contrast to OLEDs, is therefore primarily not spin dependent. As such, the subsequent discussion strictly only applies to OLEDs comprising materials of weak spin-orbit coupling, i.e., of low atomic-order number. The appeal of experimenting with spins in OLEDs is that their coherence time, i.e., the transverse spin-relaxation time $T_2$, is only weakly dependent on temperature[15]. This independence is a direct consequence of the lack of spin-orbit coupling, which leaves the carrier dynamics in the local hyperfine fields as the coherence-limiting effect[15].

In principle, magnetic resonances can be resolved down to very small frequencies of a few megahertz, limited only by the overall strength of the hyperfine interaction[5,16]. One advantage of EDMR in OLEDs over conventional EPR in radical-pair-based spin-½ systems[17] is that the sample volume can be made almost arbitrarily small. It is therefore possible to achieve high levels of homogeneity both in terms of the static field $B_0$, which defines the Zeeman splitting, and the amplitude of the resonant driving field, $B_1$, whereas at the same time penetrating the non-perturbative regime of ultrastrong drive where $B_1$ becomes comparable to $B_0$, so that the Rabi frequency approaches the Larmor frequency[18]. Such an ultrastrong-drive regime is of great interest in contemporary condensed-matter physics, although embodiments thereof have proven very challenging to find[19,20] and mostly arise in the form of ultrastrong coupling of two-level systems in resonant optical cavities[21–23].

The breakdown of the perturbative regime of OLED EDMR has previously been identified under drive conditions of $B_1 \approx 0.1B_0$, where conventional power broadening gives way to a variety of strong-drive effects[24]. Once the driving-field strength exceeds the inhomogeneous broadening of the individual spins of the pair induced by the hyperfine fields, the spins become indistinguishable with respect to the radiation and a new set of spin-pair eigenstates is formed[25]. The resonant field locks the spin pairs into the triplet configuration, in analogy to the formation of a subradiant state in Dicke's description of electromagnetic coupling of non-interacting two-level systems[26,27]. The spin-Dicke state manifests itself in EDMR by the appearance of a particular inverted resonance feature[25,28]. Experimentally, it is challenging to access this regime of strong and ultrastrong drive for the simple reason that very high oscillating magnetic field strengths have to be generated in close proximity to the OLED. Using either coils[24] or a monolithic microwire integrated in the OLED structure[18], we were previously able to probe the strong-drive regime of up to $B_1 \approx 0.3B_0$. We indeed observed an inversion of the resonance signal in the device current along with spectral narrowing occurring owing to the spin-Dicke effect[18,24]. Larger oscillating field strengths were not accessible with these earlier experimental setups. In addition, the magnitude of the $B_0$ field necessary to clearly resolve the resonance was previously limited by the inhomogeneous broadening of the resonance spectrum due to the hyperfine fields arising from the omnipresent protons[16]. Besides limiting spectral resolution, this inhomogeneous broadening also constrains the effective coupling strength of the spin states to the driving field[25]. Qualitatively speaking, the disorder increases quantum-mechanical distinguishability of the individual spins with respect to the driving field, lowering the overall degree of coherence of the incident radiation with the spin ensemble that can be achieved under resonant drive[25].

To date, there has been no formulation of the theoretical expectation of the nature of spin-dependent transitions of two weakly coupled spin-½ carriers, electron and hole, under ultrastrong resonant drive. Although a Floquet formalism to treat this problem has been put forward previously, this was only pursued in the weak-drive limit[29].

In this work, we begin by setting out a strategy for computing such transitions in the ultrastrong-drive regime using the periodic time-dependent spin Hamiltonian in the Floquet formalism. The numerically rather detailed calculations can be condensed into a diagrammatic representation of the resulting hybrid light-matter states, the spin states of the OLED dressed by the driving electromagnetic field. This approach gives us a complete computation of the EDMR magnetic resonance spectrum of an OLED under the condition of ultrastrong drive, with $B_1$ exceeding $B_0$. With substantial experimental improvements both in terms of the material used and with regard to the monolithic OLED-microwire

structure we succeed in experimentally identifying the main predicted Floquet spin states of the OLED under ultrastrong drive conditions. These states are manifested as magnetic-dipole-forbidden multiple-quantum transitions and resonances at fractional g-factors.

## Results

**Spin transitions in OLEDs under non-perturbative resonant drive.** Theoretical treatment of the spin dynamics in a strongly driven electron-hole pair beyond the perturbative regime is extremely challenging and has not been considered in detail previously. We approach this problem using quantum-mechanical Floquet theory[30]. The electron-hole spin-pair Hamiltonian, $H(t)$, is time dependent owing to the presence of a time-dependent (sinusoidal) driving field. Besides the drive, we incorporate in $H(t)$ the electron and hole spin coupling to the external field $B_0$ and the effective local internal hyperfine fields, as well as the isotropic exchange and dipolar interactions between the electron and hole spins. Note that, given the exceedingly weak spin-orbit coupling of the organic semiconductor material used in the experiments discussed in the following, the spin Hamiltonian is perfectly defined without an explicit spin-orbit term, using effective Landé g-factors instead. We direct the interested reader to a recent joint experimental and theoretical examination of the influence of spin-orbit coupling in these materials[14]. Although all these interactions are time independent, the local hyperfine fields and the dipolar interaction are taken to be random and different for different configurations. Further details of the statistical distribution of spin pairs and the numerical approach to account for this are discussed in Supplementary Notes 1.5 and 1.6.

The time-periodic character of the driving field allows us to write the Fourier decomposition of the Hamiltonian as $H_{\alpha\beta}(t) = \sum_n H_{\alpha\beta}^{(n)} e^{in\omega t}$, where $\omega$ is the angular frequency of the incident radiation. Here, and in the following, the Greek indices $\alpha$ and $\beta$ run over the four spin-pair states, the three triplets $T_+$, $T_0$, $T_-$ and the singlet $S$, which are chosen as the spin Hilbert space basis for the subsequent discussion. The Floquet dressed states $|\alpha, n\rangle$ with photon number $n = 0, \pm 1, \pm 2, \ldots$ are defined as the product of the spin Hilbert space basis state $\alpha$ and the Fourier-space basis state $|n\rangle$. The dressed states form an orthonormal basis in an infinite-dimensional Floquet Hilbert space. A change of $n$ modifies the dressed state while retaining the same spin-pair component. The time-independent, infinite-dimensional Floquet Hamiltonian $H_F$ is then defined in matrix representation as

$$\langle \alpha, n | H_F | \beta, m \rangle = H_{\alpha\beta}^{(n-m)} + n\omega \delta_{\alpha\beta} \delta_{nm}. \tag{1}$$

The Floquet approach takes advantage of the fact that $H_F$ is time independent and Hermitian, thus possessing stationary eigenvectors, $|\psi_{\alpha,n}\rangle$, and real eigenvalues, $\varepsilon_{\alpha,n}$. The Floquet Hamiltonian has a periodic structure so that a shift of the indices by the same integer $l$ changes only the diagonal matrix elements

$$\langle \alpha, n+l | H_F | \beta, m+l \rangle = \langle \alpha, n | H_F | \beta, m \rangle + l\omega \delta_{\alpha\beta} \delta_{nm}, \tag{2}$$

rendering periodicity of the eigenvectors, $\langle \alpha, n+l | \psi_{\alpha,m+l} \rangle = \langle \alpha, n | \psi_{\alpha,m} \rangle$, and eigenvalues, $\varepsilon_{\alpha,n+1} = \varepsilon_{\alpha,n} + l\omega$.

The spin-density matrix of an ensemble of spin pairs, $\rho$, satisfies the stochastic Liouville equation and is used to compute the steady-state singlet content of the spin pair, i.e., the observable responsible for the experimentally measured conductivity of the OLED. Ultimately, this observable can be expressed by the trace of the steady-state spin-density matrix, $\mathrm{tr}\tilde{\rho}_0$, where the tilde indicates the steady-state solution to the stochastic Liouville equation, which is explicitly time dependent because of the time-periodic Hamiltonian, with the index 0

referring to the time-independent Fourier component thereof. As described in detail in Supplementary Note 1.2, from this solution, we can express $\mathrm{tr}\tilde{\rho}_0$ through the following phenomenological parameters: the rate of spin-pair generation, $G$; the spin-pair dissociation rate, $r_d$; the singlet and triplet recombination rates, $r_S$ and $r_T$; and the spin-lattice relaxation time, $T_{sl}$. For convenience, we introduce a characteristic total decay rate of the pair, $w_d = r_d + r_T + 1/T_{sl}$, and define the difference in singlet and triplet recombination rates $k_r = r_S - r_T$, which determines the overall magnetic-field response of the conductivity. Using $|\psi_{\alpha,0}\rangle$ as the eigenvectors of the Floquet Hamiltonian and $\Pi_S$ as the projection operator onto the dressed singlet subspace, $\Pi_S = \sum_n |S, n\rangle\langle S, n|$, we arrive at an expression for the steady-state spin-density matrix as

$$\mathrm{tr}\tilde{\rho}_0 \simeq \frac{G}{4} \sum_\alpha \frac{1}{w_d + k_r \langle \psi_{\alpha,0} | \Pi_S | \psi_{\alpha,0} \rangle}. \tag{3}$$

Equation (3) is derived perturbatively, to leading order in the small difference of singlet and triplet recombination rates $k_r$. Note that $k_r \ll |D|, |J|$, where $D$ and $J$ are the average dipolar and exchange interactions between electron and hole spins within the pairs. Further details of this approximation are discussed in Supplementary Note 1.4.

The steady-state device current measured in an experiment is a function of the static and oscillating magnetic fields, which give rise to a change $\Delta I(B_0, B_1) = I(B_0, B_1) - I(B_0, 0)$ of the spin-dependent OLED current, i.e., the magnetic resonance. This steady-state spin-dependent device current probed in experiments corresponds to the average of $\mathrm{tr}\tilde{\rho}_0$ over all the random distributions of local hyperfine fields and dipolar couplings, i.e. the average over spatial orientations of electronic spin pairs with respect to the applied magnetic field,

$$I = I_0 \langle \mathrm{tr}\tilde{\rho}_0 \rangle, \tag{4}$$

where $I_0$ is a spin-independent factor that is determined by the conductivity of the OLED.

We utilize Eqs. (3) and (4) for the numerical calculation of the EDMR signal $\Delta I(B_0, B_1)$. The Floquet Hamiltonian $H_F$ is determined from Eq. (1), for a randomly generated configuration of local hyperfine fields and dipolar coupling. To approximately calculate the infinite-dimensional eigenvectors $\psi_{\alpha,0}$ of $H_F$ entering in Eq. (3), we truncate the infinite-dimensional Floquet Hamiltonian and the corresponding Hilbert space as described in Supplementary Note 1.3. The truncation procedure consists of restricting the photon numbers by some integer value, $N_0$, or equivalently restricting the indices $m$ and $n$ in Eq. (1) to run between $-N_0$ and $N_0$, where $N_0$ is determined by the requirements on the accuracy of the numerical procedure. The eigenvectors of the truncated Floquet Hamiltonian are used to evaluate $\mathrm{tr}\tilde{\rho}_0$ from Eq. (3). The procedure is repeated multiple times and the field-dependent current is found as the average of the numerical values computed for each random configuration according to Eq. (4).

**Diagrammatic representation of hybrid light-matter states.** In order to build up an intuitive understanding of the resonant transitions of the electron-hole spin pair emerging under strong and ultrastrong resonance drive and to differentiate the Floquet states responsible for the specific transitions, we develop a diagrammatic representation[31] of the dressed states $|\alpha, n\rangle$. For simplicity, we describe the transitions in terms of effective g-factors of resonances. Obviously, this does not relate to the g-factor of the resonant spins—these are all nearly free electrons—but to the ratio between the driving field oscillation frequency and the frequency

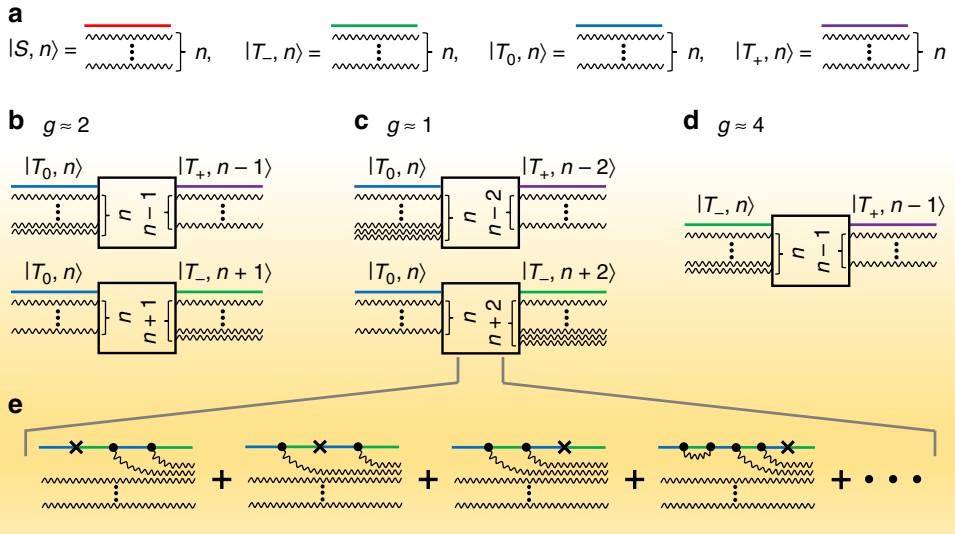

**Fig. 1 Multiple-quantum transitions of spin pairs in the singlet-triplet basis. a** The Floquet states describing the conductivity of an OLED under magnetic-resonant excitation are defined by the photon number $n$ (illustrated as wavy lines before and after the interaction) and the spin wavefunction (red, green, blue, purple). **b** The spin-½ resonance of the pair at an effective g-factor of $g \approx 2$ corresponds to a raising or lowering of $n$. **c, d** Resonances also arise at $g \approx 1$ and $g \approx 4$ owing to two-photon and half-field transitions, respectively. **e** Examples of diagrams of the two-photon transitions, where the vortices indicate the creation or annihilation of a photon (●) or spin scattering not involving a photon (×), e.g., owing to hyperfine or dipolar coupling. An infinite number of higher-order loops exists. Magnetoresistance on resonance is calculated by summation over all transitions.

corresponding to the Zeeman splitting induced by the static magnetic field $B_0$. Figure 1a illustrates the fourfold basis of the Floquet states. A resonant transition involving the creation or annihilation of a photon, shown in Fig. 1b, gives rise to an effective resonance in the overall singlet content of the pair at a g-factor $g \approx 2$. This resonance arises from a superposition of transitions depicted by an infinite number of diagrams of increasing order. Specific examples of such diagrams describing the $g \approx 1$ two-photon resonance owing to simultaneous annihilation or creation of two photons, shown in Fig. 1c, are given in Fig. 1e. The vortices of the diagrams mark transitions in spin state and occur either by photons, marked as dots, or due to magnetic scattering interactions such as hyperfine or dipolar spin-spin coupling (crosses). The entire singlet content of the spin ensemble in the steady state is computed by approximating the infinite summation over all states (see Supplementary Note 1.4). A further group of transitions (Fig. 1d) involves a change of the magnetic quantum number by $\Delta m = \pm 2$, corresponding to a half-field resonance at $g \approx 4$. Since the steady-state singlet content is given by a summation over all spin permutations, resonances at fractional g-factors will also arise as combinations of the different processes.

Even though multi-photon transitions in two-level spin systems have been discussed in the context of electron-spin resonance spectroscopy[32–34], these are not analogous to two-photon absorption of electric-dipole transitions. As each photon carries angular momentum $\ell = 1$, a $\Delta \ell = 2$ electric-dipole transition requires a final electronic orbital with different angular momentum. A spin-½ system can only undergo $\Delta \ell = 1$ transitions, however, implying that a magnetic-dipole transition by absorption of two identical photons is impossible. Instead, two-photon magnetic-dipole transitions arise with a combination of different photons[34] whose magnetic-field components have transversal and longitudinal orientation with regard to the magnetic field $B_0$.

**Computation of the EDMR spectrum as a function of driving strength.** In Fig. 2a the calculated change of spin-dependent current $\Delta I(B_0, B_1)$ as a function of Zeeman splitting $B_0$ and

driving field amplitude $B_1$ for an incident radiation frequency of 85 MHz is plotted (see Supplementary Note 1.5 for numerical details). The width of the fundamental resonance $g \approx 2$ at low driving fields is defined by the expectation value of the local hyperfine field experienced by electron or hole spins, $\Delta B_{hyp}$[25,28], as described in Supplementary Note 5. The effective g-factors of the resonant species are indicated on the left-hand field ordinate. Six distinct resonances are seen, with the amplitudes and positions of these depending on $B_1$. The six features are assigned the Floquet-state transitions marked in the diagram of spin-pair eigenstates in Fig. 2b. Here, states $|2\rangle$ and $|3\rangle$ depend on the particular nature of the intrapair interaction, i.e., dipolar and exchange coupling (see Supplementary Note 1.1). Three resonances arise from full-field $\Delta m = \pm 1$ transitions, and three from half-field $\Delta m = \pm 2$ transitions. Feature (i) is owing to the one-photon spin-½ resonance and initially undergoes power broadening, before splitting due to the AC-Zeeman effect and subsequently inverting in amplitude owing to the spin-Dicke effect[24]. The $g \approx 1$ feature (ii) results from two-photon transitions, and the $g \approx 2/3$ feature (iii) from three-photon absorption. Resonances also occur between pure triplet states and result from transitions involving either one (iv) ($g \approx 4$), three (v) ($g \approx 4/3$), or five (vi) ($g \approx 4/5$) photons. Features (i–iii) clearly move towards lower $B_0$ values with increasing $B_1$, which is a manifestation of the Bloch-Siegert shift (BSS), discussed in more detail below and in Supplementary Note 6. The BSS of the $g \approx 2$ resonance results in merging into a zero-field resonance at high $B_1$. The resonances appear self-similar, with the AC-Zeeman and spin-Dicke effects apparent in features (i–iii). Crucially, the inversion of $\Delta I$ at the onset of the spin-Dicke effect coincides with the formation of a new spin basis of the electromagnetically dressed state[24]. In this hybrid light-matter state, power broadening of the resonant transition is absent since the electromagnetic field is implicit in the state's wavefunction[25], allowing the BSS to be resolved clearly.

**Experimental probing of hybrid light-matter Floquet states in OLED EDMR.** Identifying these Floquet states experimentally in the spin-dependent transitions that control the magnetoresistance

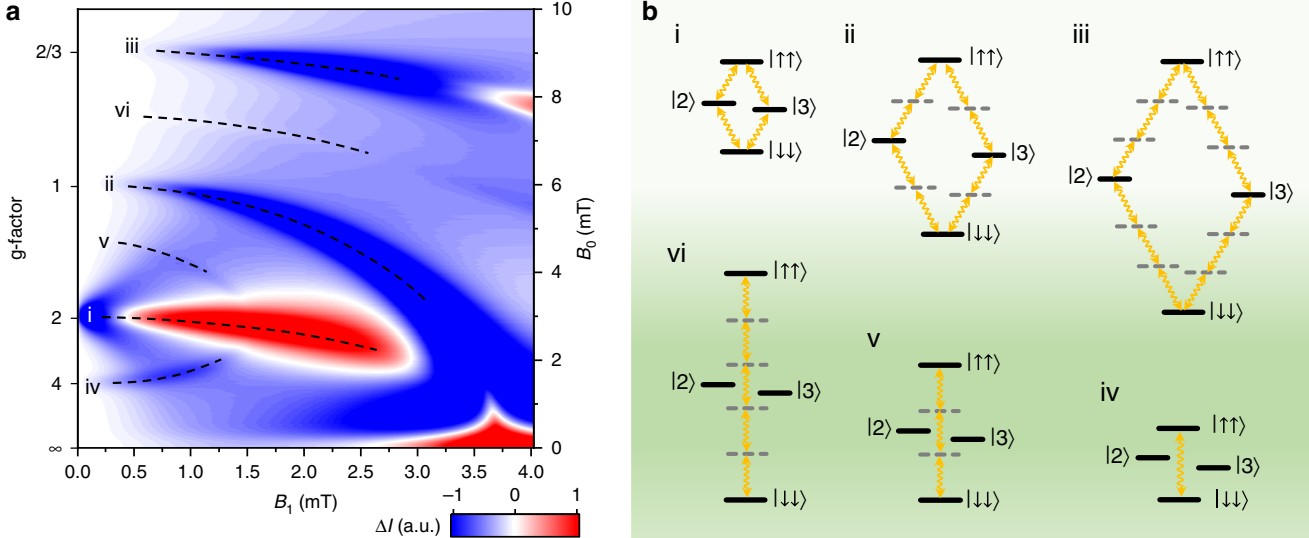

**Fig. 2 Floquet spin states in OLED magnetoresistance. a** Calculated change of spin-dependent recombination current as a function of static field $B_0$ and oscillating field $B_1$ for a frequency of 85 MHz. **b** Term diagrams of integer and fractional g-factor multi-photon transitions. (i–iii) $\Delta m = 1$ transitions; (iv–vi) $\Delta m = 2$ transitions.

of OLEDs poses two challenges. First, the effective hyperfine fields must be sufficiently small, such that the different resonances do not overlap spectrally. Hyperfine coupling broadens the resonant magnetic-dipole transition inhomogeneously, making individual microscopic spins distinguishable in terms of their resonance energy. This disorder determines the threshold field for the onset of spin collectivity[24], when each individual spin becomes indistinguishable with respect to the driving field. Previous studies of the condition of strong magnetic-resonant drive of OLEDs employed either conventional hydrogenated organic semiconductors[18,24], or partially deuterated materials with reduced hyperfine coupling strengths[24]. To maximize the resolution of the experiment, we synthesized a perdeuterated conjugated polymer, poly[2-(2-ethylhexyloxy-$d_{17}$)-5-methoxy-$d_3$-1,4-phenylenevinylene-$d_4$] (d-MEH-PPV), with 97% of the protons replaced by deuterons[35]. Second, OLEDs have to be designed with integrated microwires which generate the oscillating field[18]. The smaller the OLED pixel relative to the microwire, which narrows down to a width of 150 μm, the lower the inhomogeneity in both static and oscillating magnetic fields—at the cost of EDMR signal-to-noise ratio. The alternating current passed through the microwire generates heat, requiring careful optimization of electrical and thermal conductivity of the monolithic OLED-microwire device[18].

Figure 3 plots the change $\Delta I$ of the steady-state DC forward current $I_0 = 500$ nA under 85 MHz RF radiation, as a function of $B_0$ and the square root of the power $P$ of the applied radiation. The dominant Floquet spin state transitions identified in the calculation in Fig. 2 are resolved in the experiment and labeled correspondingly: the power-broadened $g \approx 2$ spin resonance which inverts its sign in the spin-Dicke regime with subsequent BSS (i); the two-photon transition and the corresponding BSS of the $g \approx 1$ resonance (ii); and the half-field resonance at $g \approx 4$ (iv). In principle, $\sqrt{P} \propto B_1$, although heating of the microwire at high powers will change the microwire impedance and therefore increase the uncertainty on the abscissa scale in Fig. 3 (see Supplementary Note 4). From an analysis of the experimentally observed power broadening given in Supplementary Fig. 2, we estimate that $B_1$ fields of up to 3 mT are reached, consistent with the universal scaling of EDMR amplitude dependence on $B_1$ with

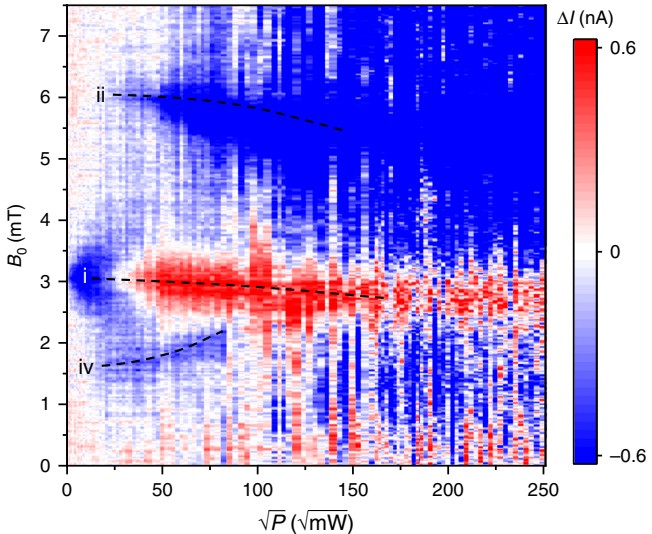

**Fig. 3 Measured change in spin-dependent OLED current $\Delta I$ as a function of driving power.** The dominant features of Fig. 2 are indicated by dashed lines. The relationship $\sqrt{P} \propto B_1$ is confirmed for $P \leq 100$ mW (see Supplementary Note 4), but, due to heating-induced resistivity changes of the microwire generating the resonant field, the uncertainty on the abscissa scale increases with higher $P$.

hyperfine field strength as shown in Supplementary Fig. 6. Note that, because the measurements were actually carried out before the calculations were completed, unfortunately, we did not probe the $B_0$ field region above 7.5 mT, as there was no reason to anticipate additional features appearing there. We will leave experimental exploration of this interesting field region for future studies.

A simple intuitive rationalization of the intriguing two-photon transition can be formulated. Two-photon transitions between the up and down state of a spin-½ species are dipole forbidden. In analogy to light waves, the resonant radiation can be described in the photon picture. A photon has an angular momentum quantum number of 1 and can assume one of the two projection

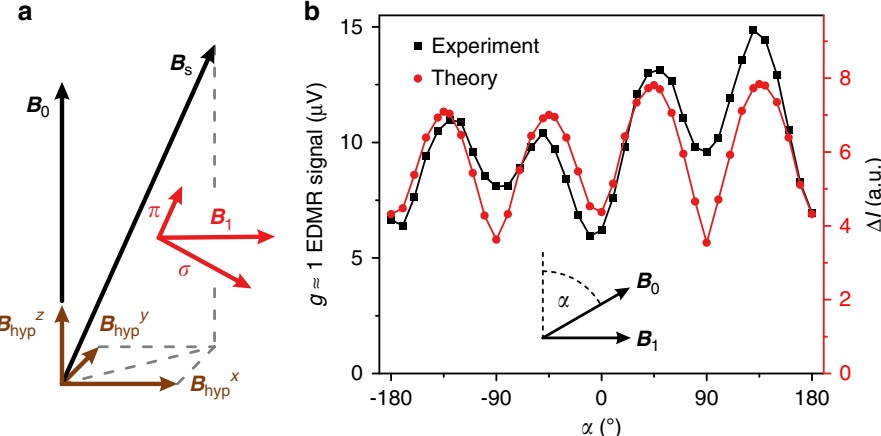

**Fig. 4 Intuitive rationalization of the two-photon transition and the angular dependence of the two-photon $g \approx 1$ EDMR signal. a** Sketch of the magnetic-field configuration acting on a spin within a spin pair. Oscillating magnetic-field components (red arrows) parallel ($\pi$) and perpendicular ($\sigma$) to the effective local quantizing static magnetic field $B_s$ arise from the decomposition of $B_1$ with respect to the total static magnetic field that results from the superposition of $B_0$ and $B_{hyp}$. **b** Angular dependence of the two-photon EDMR resonance measured at $f = 22$ MHz (black) for the d-MEH-PPV devices. The experimental results match the $\sin 2\alpha$ dependence predicted by theory (red). Lines are a guide to the eye.

states of either $m = +1$ or $-1$ relative to the direction of propagation. These so-called $\sigma^+$ and $\sigma^-$ photons can be associated with the right and left-hand circularly polarized fields. In a $\sigma$-photon state, where the direction of propagation is along the quantization axis, the AC magnetic-field component $B_1$ is perpendicular to $B_0$. The absence of a photon state with $m = 0$ can be traced back to a photon having zero mass. However, a linear AC field $B_1$ in the direction of quantization, $B_1 \parallel B_0$, is referred to as a $\pi$-photon propagating perpendicular to the quantization axis. These unconventional $\pi$-photons feature in atomic spectroscopy but are also discussed within a semi-classical picture of magnetic resonance[2,34], and can be thought of as photons of total angular momentum zero. Two-photon transitions are only possible with a combination of $\sigma$ and $\pi$-photons. As the excitation scheme employed here would appear to only involve $\sigma$-type photons, it is not immediately obvious how angular momentum is conserved to give rise to the strong two-photon resonances observed in theory and experiment.

Figure 4a provides an intuitive picture for the emergence of the two-photon transition. Transversal oscillating magnetic-field components are orthogonal to $B_0$, but the latter is superimposed with local static isotropic hyperfine fields. This superposition leads to the effective static magnetic field, $B_s$, tilted from the $z$ axis by a small but finite angle. Thus, the superposition of $B_0$ and the local hyperfine fields results in a total static field $B_s$ parallel to one component of the oscillating magnetic field, which is sufficient to enable the $\sigma - \pi$ two-photon transition conserving angular momentum. To test this picture, in a separate experimental setup described in the Methods section, we probed the angular dependence of the $g \approx 1$ resonance. The complete quantum-statistical simulation of the effect using the Floquet ansatz described above predicts a $\sin 2\alpha$ dependence, in agreement with the geometric arguments forward above. The characteristic features of the prediction from theory are matched by the experimental data as shown in Fig. 4b. To allow the angle to be swept, these experiments were performed at a very low $B_0$ of 780 µT corresponding to a resonance frequency of 22 MHz.

Two-photon resonances arise both under parallel and perpendicular excitation. This observation can be explained by the illustration in Fig. 4a: photons of both longitudinal ($\pi$) and transversal ($\sigma$) polarizations exist even for parallel or perpendicular field configurations. The lower $B_0$, the greater the contributions of the transversal hyperfine fields and the influence of dipolar

coupling of spins within the spin pairs, which enable the resonances under fully parallel and perpendicular excitation. We note that a slight asymmetry in the angular dependency can be attributed to additional resonant contributions of the second harmonic of the RF signal generated by the RF amplifier. This second harmonic can be removed by filtering the incident RF radiation using a low-pass filter. Alternatively, we replicate this asymmetry in the calculation in Fig. 4b by introducing a second harmonic resonant contribution amounting to ~0.01 $B_1$. The additional slight asymmetry in the measurement between, e.g., 45° and 135° is also observed for the fundamental resonance (not shown). This deviation presumably arises from the fact that the RF radiation generated by the stripline is not perfectly linearly polarized.

## Discussion

We conclude that the Floquet ansatz presented here to solve the time-dependent Hamiltonian of a two-level system in the ultrastrong-drive regime provides both an accurate and surprisingly intuitive representation of the complex spin excitations arising under these conditions. These include the spin-Dicke state, which manifests a strong BSS, along with dressed light-matter states, which support dipole-forbidden multiple-quantum transitions. Spin-dependent recombination rates between paramagnetic charge-carrier states in OLEDs offer a remarkably versatile testbed to probe this regime of ultrastrong coupling, visualizing directly the predicted multiple-quantum transitions and fractional g-factor resonances—to our knowledge, better than other known quantum systems at room temperature. The identification of the BSS at room temperature is particularly clear in these solid-state devices compared to any other spectroscopic probe of this phenomenon outside of nuclear magnetic resonance[36–38]. Similarly, direct signatures of hybrid light-matter Floquet states are observed here to influence spin-dependent OLED currents, much like the Floquet states reported in photo-electron spectroscopy of topological materials[39].

The theoretical results of Fig. 2 provide motivation to extend the phase space of the experiment to even higher ratios of $B_1/B_0$. Although experimentally challenging, this is not entirely unfeasible, and may, for example, be conceivable by using superconducting stripline resonators[9]. It would be particularly exciting to experimentally identify the spin-Dicke state, which is reflected by the inversion and narrowing of the resonance, in the three-photon and two-photon transitions (iii) and (ii)

predicted in Fig. 2. The latter is especially interesting as the BSS appears to induce a zero-field resonance, i.e., a resonance feature for $B_0 \rightarrow 0$ mT and $B_1 \approx 3.7$ mT.

Although we refrain at present from speculating too much on possible technological applications of our study, we do note that a clear feature in theory and experiment is the demonstration that power broadening is suppressed when the spin-Dicke state is formed: the one-photon resonance feature (i) power-broadens linearly with driving field strength $B_1$ when $B_1$ is below the Dicke regime, whereas after inversion of the resonance sign (marked in red in Figs. 2 and 3) broadening of the inverted feature with a further increase in power is minimal. This absence of power broadening implies that the hybrid light-matter state, the spin-Dicke state[25], appears to be protected against power broadening precisely because it is a hybrid state. The consequence of this protection should be a dramatically enhanced coherence time, which in turn may turn out to be useful in quantum magnetometry applications. Measuring this increased coherence time in pulsed experiments[40] at very high $B_1$ could potentially confirm this hypothesis. Such experiments would require coherent spin manipulation by RF pulses, which are shorter in duration than the RF wave period at the very low Larmor frequencies used here, i.e., subcycle magnetic-resonant coherent control, which is technically feasible with modern pulse synthesizers.

Finally, the work presented here also suggests exciting challenges for materials chemists. The linewidth of the resonances in the calculated spectra of Fig. 2 is determined primarily from the finite inhomogeneous broadening arising from the residual hyperfine field strengths of the deuterated conjugated polymer. By careful design of materials, it may be possible to reduce hyperfine coupling even further, for example, by extending the π-electron system in polycyclic aromatic hydrocarbons to lower the interaction between electronic and nuclear magnetic moments. With such advances, it will be possible to resolve resonances at even lower $B_0$ fields and therefore reach even higher effective ratios of $B_1$ to $B_0$.

## Methods

Details of the computational procedure, including the modifications made to the conventional spin-pair model of spin-dependent recombination in an OLED[28,41], are given in Supplementary Note 1.5 along with a list of the model parameters used.

**Device fabrication**. In order to establish and detect ultrastrong electron-spin magnetic resonance drive conditions, we developed a new monolithic OLED device structure with an integrated RF microwire. These small circular (57 μm diameter) single-pixel OLEDs allow for bipolar charge-carrier injection using electron and hole injector layers on top of an electrically and thermally separated thin-film wire capable of producing RF magnetic fields $B_1$ when connected to an RF source. The wire itself runs across the substrate and is shrunk down to 150 μm width beneath the OLED pixel. Details of the layer sequence necessary to achieve electrical and thermal decoupling are given in ref. [18]. Further details can be found in Supplementary Note 2. The active polymer, d-MEH-PPV, was dissolved in toluene at a concentration of 4.5 g/L at a temperature of 50–70 °C and deposited by spin-casting at 800 rpm, as described in ref. [35]. The layer was sandwiched as a 100 nm thin film between a Ca/Al stack for electron injection and a TiO$_2$ (3 nm)/Au (80 nm)/Ti (10 nm) layer covered with PEDOT:PSS (Ossila Al 4083) for hole injection. The structure is comparable to that used in previous studies of the strong-drive regime, where a different π-conjugated polymer with much larger resonance linewidths was used[24]. For the study presented here, the active-layer material, the layer stack, and the pixel device geometry were redesigned and optimized in order to allow for larger $B_1/B_0$ ratios to be reached. The monolithic single-pixel OLED device was made following a preparation and deposition protocol as illustrated in Jamali et al.[18] with the following changes: (i) the active polymer layer where the electron-hole pair recombination takes place was nominally 100 nm thick, consisting of spin-coated perdeuterated d-MEH-PPV; (ii) the sample template preparation included placing the silicon wafer in a dry thermal oxidation furnace at 1000 °C for 78 min, producing a 50 nm SiO$_2$ layer on top of the Si wafer for insulation and better adhesion of the subsequent layer (amorphous SiN); and (iii) an entirely different lateral layout of the templates (cf. Supplementary Fig. 1a) was used compared to that described in Jamali et al.[18], providing larger separation between

the electrical contacts of the thin-film wire RF source and the device electrodes in order to avoid cross-talk at high RF powers and minimize heating of the OLED during the room temperature measurements. A photograph of the pixel device template is shown in Supplementary Fig. 1a with the pixel located at the image center. An example of the current-voltage characteristics of the device at room temperature is also given in Supplementary Fig. 1b.

**Ultrastrong-drive EDMR spectroscopy**. The OLED was subjected to a static magnetic field $B_0$ and irradiated with RF radiation with in-plane amplitude $B_1$ using an Agilent N5181A frequency generator and an ENI 510 L RF power amplifier as illustrated in Supplementary Fig. 1a. At the same time, a steady-state electric current was induced with a $V = 2.8$ V bias using a Keithley voltage source. A Stanford Research (SR570) current amplifier was used to detect changes $\delta I$ of the steady-state forward current of $I_0 = 500$ nA with and without the RF radiation applied. Measurements of $\Delta I(B_0, \sqrt{P})$ as a function of $B_0$ and the square root of the applied microwave power $P$ (which is proportional to $B_1$) were then obtained, as shown in Fig. 3. Further details of the measurement procedure and the determination of the $\sqrt{P}$ to $B_1$ conversion factor through power broadening are given in the Supplementary Notes 3–7.

**Angle-dependent EDMR spectroscopy of the two-photon transition**. The angle-dependence measurements of the two-photon resonance were performed in a separate low-field setup with different OLED samples, described in detail in refs. [5,42]. These OLEDs were much larger, with a pixel area of 3.5 mm². RF excitation generated by an Anritsu MG3740A signal generator and amplified by a HUBERT A 1020 RF amplifier was applied to the sample through a coplanar stripline designed to match a 50 Ω impedance. In contrast to the measurements on the monolithic OLED-microwire structures, these experiments were performed under constant-current conditions with $I = 250$ μA applied by a Keithley 238 source-measure unit, so that the EDMR signal corresponds to a voltage measurement. Lock-in detection with a Stanford Research Systems SR830 DSP lock-in amplifier and 100% modulation of the RF amplitude at a frequency of 232 Hz was employed. We used a 3D array of Helmholtz coils (Ferronato BH300-3-A), each supplied by a CAEN ELS Easy-Driver 0520. This allowed us to perform sweeps of $B_0$ from −2 to +2 mT with an arbitrary orientation of the field vector. Owing to the limited magnetic-field range of the 3D Helmholtz-coil set, we resorted to a lower excitation frequency of 22 MHz with $B_1 \approx 200$ μT.

## Data availability

The raw data that support the plots within this paper and the other findings of this study are available from the corresponding authors upon reasonable request.

## Code availability

The Fortran code used for the numerical simulations discussed in this paper is available from the corresponding authors upon reasonable request.

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

## Acknowledgements
This work was supported by the US Department of Energy, Office of Basic Energy Sciences, Division of Materials Sciences and Engineering under Award #DE-SC0000909. The synthesis of the perdeuterated monomer took place at the Australian National Deuteration Facility, which is partly funded by NCRIS, an Australian Government initiative. P.L.B. is an Australian Research Council Laureate Fellow and the synthesis work was supported in part by this Fellowship (FL160100067). J.M.L. and S.B. acknowledge funding by the Deutsche Forschungsgemeinschaft (DFG, German Research Foundation)—Project-ID 314695032—SFB 1277.

## Author contributions
S.J. designed the high-power microwire OLED structure and performed magnetic resonance spectroscopy, with help from H.M., A.N., and H.P. V.V.M. developed the Floquet simulation code and performed all calculations discussed in the paper. T.G., S.B., and S.M. performed additional supporting experiments at low fields. P.L.B. led the synthesis of the perdeuterated conjugated polymer with support from D.M.S., A.E.L., and T.A.D. J.M.L. and C.B. conceived and supervised the project, and wrote the paper with input from all authors.

## Funding

## Competing interests
The authors declare no competing interests.
