## [Peer Review File · Nature Communications]

Reviewers' Comments:

Reviewer #1:

Remarks to the Author:

I believe that the text of the paper is now much clearer in its scope. It also provides a well argued vision for a compelling frontier in modern optical physics.

The manuscript could have a tongue-in-cheek title of the type "A manifesto for non-perturbative Floquet spin OLEDs" and is perhaps slightly verbose without extensive experimental evidence.

However, in the aggregate I am inclined to recommend publication of this interesting and engaging paper in *Nature Communications*. The writing is crisp and the discussion very interesting and I am pretty certain that this work will be well read and referenced.

Reviewer #3:

Remarks to the Author:

Studying the comments of the two referees, I come to the conclusion that these referees are in principle enthusiastic about the paper. The referees do have a significant amount of critical remarks, to which the authors provide lengthy replies. My impression is that, despite the fact that I find these replies sometimes unnecessarily lengthy and not to the point, they generally do answer the main points of criticism. It is of course up to these referees themselves to judge if they find the replies satisfactory at every specific point.

Like these two referees, I am generally enthusiastic about the paper. The subject of Floquet states in OLEDs is undoubtedly scientifically very interesting. The authors develop a theoretical method to describe them that is inspired by other work ([29],[31]), but contains several exciting new aspects. The experimental verification is a great achievement, pushing magnetic resonance engineering to its limits. In my opinion, the combination of these two aspects warrants publication in a journal like *Nature Communications*.

Obviously, I have also many questions and remarks. However, I understand that a long list of critical remarks, in addition to those of the first two reviewers, will lead to further complications and delays for the authors, which is not desired at this stage. It is important that the scientific community is now quickly informed about this interesting work. The authors have a reputation in the field as outstanding scientists, being very knowledgeable in the fields of organic electronics and magnetic resonance, and I trust that they will have good answers to various detailed questions I still have myself.

There are, however, two issues that are too bothering for me and that should in my view be resolved before acceptance:

1) The authors talk on p. 17 about a separate experiment where they probe the angular dependence

of the $g \approx 1$ resonance. They then start describing this experiment and end by saying that "We refrain from presenting these data here and will revisit the angular dependence in a future in-depth analysis." I think this is unacceptable. In their own words, these data "offer an additional test of the theory". These data are therefore vital information for the reader and should be presented.

2) The B_0 -axis in the experiments stops at 7.5 mT, whereas in the theory it stops at 10 mT. The authors say that "for experimental reasons, B_0 was limited to 7.5 mT", without mentioning these reasons. This is puzzling, because it is hard to think of a good experimental reason why it would be troublesome to go beyond a modest field of 7.5 mT. It is also disappointing, because the theory still predicts interesting Floquet state features between 7.5 and 10 mT. It is strange that this is not discussed, because the authors do discuss other interesting features they would like to study, such as the zero-field resonance at high B_1 . The authors surely do not want to make the impression that they actually can go beyond 7.5 mT, but do not observe the predicted Floquet state features. A complete answer is required here.

Response to reviewers.

We thank the reviewers for their thoughtful feedback on our manuscript. This time round, we will avoid making the “*replies sometimes unnecessarily lengthy*” (Rev. #3).

Rev. #1

“The manuscript could have a tongue-in-cheek title of the type "A manifesto for non-perturbative Floquet spin OLEDs" and is perhaps slightly verbose without extensive experimental evidence.

However, in the aggregate I am inclined to recommend publication of this interesting and engaging paper in Nature Communications. The writing is crisp and the discussion very interesting and I am pretty certain that this work will be well read and referenced.”

We leave it to the editor to decide on the title. We are not sure the suggested title is an improvement.

We are not entirely sure we understand how the writing is both “crisp” and “verbose” at the same time, but we take both as compliments.

Rev. #3

“1) The authors talk on p. 17 about a separate experiment where they probe the angular dependence of the $g \approx 1$ resonance. They then start describing this experiment and end by saying that “We refrain from presenting these data here and will revisit the angular dependence in a future in-depth analysis.” I think this is unacceptable. In their own words, these data “offer an additional test of the theory”. These data are therefore vital information for the reader and should be presented.”

OK, we agree. These data were hidden at the end of the SI in the original submission. They are indeed significant and quite non-trivial, but we felt that they will be lost to a broader audience in the SI so we took it out in the last iteration with the intention to publish it separately.

We have reinserted the plot now, moving it to the main text, along with the intuitive rationalization of the two-photon transition (Section S1.7 of the SI), as a new Figure 4. The text and figures were previously in the SI.

“2) The B_0 -axis in the experiments stops at 7.5 mT, whereas in the theory it stops at 10 mT. The authors say that “for experimental reasons, B_0 was limited to 7.5 mT”, without mentioning these reasons. This is puzzling, because it is hard to think of a good experimental reason why it would be troublesome to go beyond a modest field of 7.5 mT. It is also disappointing, because the theory still predicts interesting Floquet state features between 7.5 and 10 mT. It is strange that this is not discussed, because the authors do discuss other interesting features they would like to study, such as the zero-field resonance at high B_1 . The authors surely do not want to make the impression that they actually can go beyond 7.5 mT, but do not observe the predicted Floquet state features. A complete answer is required here.”

The referee is absolutely correct. But, unfortunately, experimental physics does not always work the way that would seem most obvious. As we say in the introduction (and in the previous response), the work was motivated by our 2017 Nano Lett. paper. Based on that paper, we got the chemist to make us the perdeuterated compound, and carried out the measurements. We then developed the theory to understand these, since the experimental results were so unexpected. Before we had the theory, there was no reason to extend the B_0 range in experiment to go beyond 7.5 mT. We were very much expecting to be able to go back to the lab and extend the measurement range after submitting the paper last year, and hoped to have an extended dataset to put in the final paper, but life got in between:

- The first-author student left and was not able to return to repeat the measurements because of complications related to the current administration’s immigration policies.
- We had a water leak in the lab, which closed it down for several months last year.
- And then The Pandemic put a halt to everything. Even though we have now trained a new student, we need cleanroom access to fabricate the device templates in Fig. S2.1. This is not possible at the moment.

So, it is what it is. Obviously, we cannot put it like this in the paper (although everybody is facing unusual challenges at the moment).

The elegant solution would be to cut back the range of theory to match the experimental range, but this seems rather unwise because we would be hiding valuable scientific results. But as the referee writes, it does look as though we are hiding something by limiting the experimental range. So let’s just be frank about it. We have added the following sentence on p. 17:

“Note that, because the measurements were actually carried out before the calculations were completed, unfortunately, we did not probe the B_0 field region above 7.5 mT since there was no reason to anticipate additional features appearing there. We will leave experimental exploration of this interesting field region for future studies.”

Reviewers' Comments:

Reviewer #1:

Remarks to the Author:

I am happy to support publication. For the record, the title was not intended as a real suggestion, but more as a comment on the forward looking scope of the paper.

Reviewer #3:

Remarks to the Author:

The two issues that were bothering me before I could recommend acceptance of the paper have been resolved by the authors. I am happy to now recommend acceptance.